# Cardioprotection Attributed to Aerobic Exercise-Mediated Inhibition of ALCAT1 and Oxidative Stress-Induced Apoptosis in MI Rats

**DOI:** 10.3390/biomedicines10092250

**Published:** 2022-09-11

**Authors:** Niu Liu, Yingni Zhu, Wei Song, Wujing Ren, Zhenjun Tian

**Affiliations:** 1School of Physical Education, Weinan Normal University, Weinan 714099, China; 2College of Physical Education and Sports, Beijing Normal University, Beijing 100875, China; 3Institute of Sports and Exercise Biology, School of Physical Education, Shaanxi Normal University, Xi’an 710119, China

**Keywords:** aerobic exercise, myocardial infarction, ALCAT1, apoptosis

## Abstract

Cardiolipin (CL) plays a pivotal role in mitochondria-mediated apoptosis. Acyl-CoA: lysocardiolipin acyltransferase 1 (ALCAT1) can accelerate CL reactive oxygen production and cause mitochondrial damage. Although we have demonstrated that aerobic exercise significantly reduced ALCAT1 levels in MI mice, what is the temporal characteristic of ALCAT1 after MI? Little is known. Based on this, the effect of exercise on ALCAT1 in MI rats needs to be further verified. Therefore, this paper aimed to characterize ALCAT1 expression, and investigate the possible impact of exercise on ALCAT1 and its role in fibrosis, antioxidant capacity, and apoptosis in MI rats. Our results indicated that the potential utility of MI increased ALCAT1 expression within 1–6 h of MI, and serum CK and CKMB had significant effects in MI at 24 h, while LDH exerted an effect five days after MI. Furthermore, ALCAT1 expression was upregulated, oxidative capacity and excessive apoptosis were enhanced, and cardiac function was decreased after MI, and aerobic exercise can reverse these changes. These findings revealed a previously unknown endogenous cardiac injury factor, ALCAT1, and demonstrated that ALCAT1 damaged the heart of MI rats, and aerobic exercise reduced ALCAT1 expression, oxidative stress, and apoptosis after MI-induced cardiac injury in rats.

## 1. Introduction

Cardiovascular disease (CVD) is the leading cause of death globally, accounting for 17.3 million deaths per year, and nearly 800,000 people in the United States die of heart disease, stroke, and other CVDs every year [1]. Myocardial ischemia and hypoxia after myocardial infarction (MI) initially lead to oxidative stress injury, followed by cardiomyocyte apoptosis and fibrosis, and ultimately heart failure [2]. Apoptosis is one of the main pathways of myocardial cell loss after MI [3]. There is well-documented evidence that appropriate exercise training not only reduces the risk of cardiovascular diseases but also provides direct endogenous cardioprotection against various conditions, including MI [4,5,6,7,8,9,10,11,12,13]. However, the mechanisms underlying the cardioprotective effects of exercise are still unclear.

Cardiolipin (CL), an important inner membrane phospholipid of the mitochondria and sarcoplasmic reticulum, plays a pivotal role in the integrity of mitochondrial structure and mitochondria-mediated apoptosis [14,15]. Cytochrome C (Cyt C) is anchored to the mitochondrial inner membrane by CL, which can promote electron transfer in the mitochondrial inner membrane and prevent the release of Cyt C during apoptosis. Recently, growing evidence has shown that a decrease in CL content leads to mitochondrial-induced apoptosis [16,17,18,19]. Even more importantly, acyl-CoA: lysocardiolipin acyltransferase-1 (ALCAT1) is a key enzyme that regulates CL metabolism and mitochondrial function and has a remodeling effect on the CL branch chain structure. CL remodeling mediated by ALCAT1 can accelerate CL reactive oxygen production and cause mitochondrial damage [20,21,22]. CL remodeling by ALCAT1 leads to the synthesis of CL with an aberrant acyl composition commonly found in heart diseases, including depletion of linoleic acid and enrichment of docosahexaenoic acid (DHA) content. It has been reported that upregulation of ALCAT1 expression in myocardial cells induces oxidative stress and mtDNA damage, whereas silencing ALCAT1 inhibits overproduction of reactive oxygen species (ROS) and alleviates mitochondrial dysfunction [17].

Exercise is a strong modulator of oxidative stress. Recent studies have shown that eight weeks of aerobic exercise can increase the hepatic proportion of CL in high-fat-diet rats [23], while low-intensity aerobic exercise can significantly increase the myocardial CL content and protect myocardial function in rats with spontaneous hypertensive heart failure [24]. Moreover, it has been reported that regular exercise elevates ROS production to a level that may induce tolerable damage, which in turn promotes cellular antioxidant systems, stimulating oxidative damage repair systems [25]. These findings indicate that the exercise-induced release of CL may play a role in exercise-mediated cardiovascular benefits. Moreover, ALCAT1 can regulate CL remodeling. Consequently, there is a crucial need to explore the functional effects of exercise-derived CL, especially the role of ALCAT1 in exercise-induced cardioprotection.

Although, we have demonstrated, in MI mice, that six weeks of aerobic exercise can reduce myocardial oxidative stress and apoptosis and ultimately improve cardiac function by inhibiting the high expression of myocardial ALCAT1 [26], what are the temporal characteristics of ALCAT1 changes in MI rats? The literature lacks answers. On the basis of previous experiments in mice, the effects of four-week aerobic exercise on myocardial and circulating ALCAT1 in MI rats need to be further verified. Therefore, this paper mainly discusses the local and circulatory characterization of ALCAT1, and investigates the role of four-week aerobic exercise in reducing local and circulating levels of ALCAT1 and oxidative stress, and improving myocardial function in MI rats.

## 2. Materials and Methods

### 2.1. Animals

As illustrated in Figure 1 healthy male Sprague-Dawley rats (200 ± 5 g, 2 months of age, *n* = 130) were obtained from the Experimental Animal Center of Xi’an Jiaotong University (China; qualified production number: SCXK (Shaan) 2016-003). All animal protocols were performed in accordance with the Review Committee for the Use of Human or Animal Subjects of Shaanxi Normal University. All animals in this study were randomly divided into two parts. The first part aimed to characterize ALCAT1 expression, which was randomized into 7 groups: CTL (normal control, *n* = 10), S (sham-surgery, *n* = 10), MI-1 h (1 h after myocardial infarction, *n* = 10), MI-6 h (6 h after myocardial infarction, *n* = 10), MI-12 h (12 h after myocardial infarction, *n* = 10), MI-24 h (24 h after myocardial infarction, *n* = 10), and MI-5d (5 days after myocardial infarction, *n* = 10) groups, which were maintained throughout the study duration. Three animals in each group were used in Masson’s trichrome staining and PCR, respectively; the remaining 4 rats were used for Western blot. The other part was used to investigate the protective effect of exercise in MI rats. The experimental animals were randomly divided into 5 groups: the normal control (CTL, *n* = 12), sham surgery (S, *n* = 12), myocardial infarction (MI, *n* = 12), control + exercise (EX, *n* = 12), and myocardial infarction + exercise (MI + EX, *n* = 12) groups. All groups in the second part were sampled 4 weeks after MI operation. The rats were housed in a temperature-controlled (21 °C) facility with a 12:12 h light: dark cycle and were housed five per cage with free access to water and standard laboratory chow. The study protocols were approved by the Ethics Committee of the Institute of Sports and Exercise Biology, Shaanxi Normal University (ethical review acceptance No: 20190918001). All experiments were performed in accordance with ARRIVE guidelines.

Myocardial infarction surgery: After fasting for 12 h, rats were anesthetized with pentobarbital sodium (30 mg/kg body weight), fixed supine on the small animal operating table, and then were ventilated with a rodent respirator (DW 300), while the electrocardiogram (ECG) was monitored throughout the operation. To establish a myocardial infarction (MI) model, after the left thoracotomy in the fourth and fifth intercostal space, the left anterior descending artery (LAD) was permanently ligated 2 mm below the left auricle with a depth of 0.3–0.5 mm using 5-0 silk thread under the microscope, and the chest wall was then closed. The surgical wounds were repaired, and the rats were returned to their cages. The operation was considered successful when ST segment elevation was observed on ECG. Sham-operated control rats underwent the same procedure without LAD ligation. None of the animals died following the operation throughout the entire study.

Exercise protocol: The aerobic training protocol was modified from a previously published procedure. One week after surgery, male Sprague-Dawley rats in the MI + EX group and EX group were placed on a motorized rodent treadmill (Model ZH-PT, Anhui Zhenghua Technology Co., Hefei, China) once daily for 5 days per week for 4 weeks. In detail, the training protocol consisted of 10 m/min for 10 min, and then, at 13 m/min for 10 min, ending with 16 m/min for 40 min. The same moderate intensity was used for all exercise groups (Figure 1).

### 2.2. Myocardial Function Assessment (Hemodynamics)

Cardiac function was determined by the hemodynamic index. After the 4-week treadmill exercise, all rats were anesthetized with pentobarbital (30 mg/kg). Meanwhile, a catheter with pressure transducer was intubated reversely from the right carotid artery to the left ventricle and data were acquired in succession by using the polygraph physiological data acquisition system (PowerLab 8/30, ML870; AD Instruments, Sydney, Australia), which quantified left ventricle systolic pressure (LVSP), left ventricle end-diastolic pressure (LVEDP), and time derivatives of the pressure change during left ventricular contraction (+dp/dt) and relaxation (−dp/dt). All rats were euthanized after the hemodynamic measurements and the heart tissue samples were collected for further histological, molecular, or both, biological analyses.

### 2.3. Myocardial and Serum Assays

The levels of oxidative stress markers and myocardial infarct-related enzymes in myocardial tissue and serum were measured using the assay kits (Beyotime Institute of Biotechnology, China). After the hemodynamics test, whole blood samples of animals were obtained from their abdominal aorta without anticoagulant, and stored at 37 °C for 4 h. After being centrifuged at 1000 rpm for 1 min, serum was removed to a new EP tube and stored at −20 °C for kit assays. Ventricular tissue samples from the peri-infarcted zone stored at −80 °C were homogenized in saline, centrifuged for 10 min at 1000 rpm, and the tissue homogenate supernatant was transferred to another new EP tube, stored at −20 °C for the assays, that is, total antioxidant capacity (T-AOC), glutathione peroxidase (GSH-PX) levels, superoxide dismutase (SOD) levels, catalase (CAT) levels, and malondialdehyde (MDA) levels were determined. Furthermore, the activity of myocardial infarct-related enzymes, creatine kinase (CK) levels, creative kinase MB (CKMB) levels, lactate dehydrogenase (LDH) levels, and brain natriuretic peptide (BNP) levels were detected with kits. The absorbance was read at 240 nm (CAT), 340 nm (CK, CKMB), 412 nm (GSH-PX), 450 nm (SOD, LDH, BNP), 532 nm (MDA), and 593 nm (T-AOC), according to the manufacturer’s instructions, respectively. In addition, cardiac caspase-3 activity was determined by using a caspase-3 colorimetric assay kit (Nanjing Jiancheng, China) following the manufacturer’s instructions. Tissue samples were lysed in a lysate mixture (caspase lysis buffer: assay buffer = 1:9), homogenized in a glass homogenizer, and transferred to a 1.5 mL centrifuge tube. The mixture was placed in an ice bath at 37 °C for 5 min for cracking, centrifuged (10,000 rpm) for 10 min at 4 °C, and the tissue homogenate supernatant was transferred to a cold EP tube. With addition of other reagents, the mixture was measured with the absorbance value at 405 nm.

### 2.4. Quantitative Real-Time PCR

Total RNA was isolated from 100 mg myocardial samples using TRIzol reagent (Takara, Japan). The RNA concentration and integrity were assessed. Five hundred nanograms of RNA was used for cDNA synthesis with a Takara Prime Script RT reagent kit. RNA samples were treated with DNase before reverse transcription to avoid genomic DNA contamination in the PCR samples. An aliquot of each cDNA sample was loaded in a Light Cycler quantitative reverse transcription polymerase chain reaction (RT-PCR) system (Takara, Japan) to determine the mRNA levels of certain genes (6–10). The sequence of the primers used to detect these genes was as follows: lclat1 F: 5′ TTGCCTCAAATCCAGTCTCA3′; lclat1 R: 5′TGGCTCTTATCA TCCT TCCAC3′; gapdh F: 5′ CAGTGCCAGCCTCGTCTCAT3′; and gapdh R: 5′AGGCCATCCACAGTCTT C 3′. The results were quantified as Ct values, with Ct being defined as the threshold cycle of polymerase chain reaction at which the amplified product was first detected. GAPDH was employed as an endogenous reference.

### 2.5. Western Blot Analysis

Total protein was extracted from tissue from the peri-infarct zone of rats using a tissue total protein extraction kit (Amresco, Radnor, PA, USA). Heart tissue samples (100 mg) were lysed with lysis buffer (RIPA: PMSF: phosphatase inhibitor = 100:1:1). After homogenization, the lysates were centrifuged, and the proteins were separated by electrophoresis on 8–10% SDS-PAGE gels and then transferred onto a NC membrane (300 mA, 1.5 h). A prestained protein marker was used to confirm transfer efficiency. To save antibodies, the NC membrane was stained with Ponceau S and cut into bands according to the molecular weight of the target protein. After being blocked with 5% milk, the immunoblots were probed with anti-ALCAT1 (1:1800; Sigma, St. Louis, MO, USA), anti-P53 (1:450; Boster, Wuhan, China), anti-Bcl-2 (1:500; Bioworld, Irving, TX, USA), anti-Bax (1:600; Bioworld, Irving, TX, USA), anti-Cyt C (1:4000; Cell Signaling Technology, Danvers, MA, USA), anti-SOD1, anti-SOD2 (1:2500; Genetex, Irvine, CA, USA), anti-brain natriuretic peptide (BNP, 1:1000; Abcam, Cambridge, UK), anti-atrial natriuretic factor (ANF; 1:3000; Genetex, Irvine, CA, USA), and anti-GAPDH (1:5000; Genetex, Irvine, CA, USA) antibodies overnight at 4 °C, and then incubated with the corresponding secondary antibodies (1:8000; Jackson Immuno Research Laboratories, West Grove, PA, USA) at room temperature for 1 h. GAPDH (1:8000; Genetex, Irvine, CA, USA), as a housekeeping protein, was used as a loading control. Subsequently, the bands were visualized with ECL reagent (Millipore, Burlington, MS, USA). Quantitative assessment of the optical density of each Western blot band was performed by the Image Processing and Analysis function of Java v1.48 (Wayne Rasband, National Institutes of Health).

### 2.6. Masson’s Trichrome Staining

To visualize and measure collagen deposition, heart tissue sections were stained with Masson’s trichrome stain according to standard methods. Briefly, myocardial tissue sections were cut at a thickness of 5 μm and placed on standard microscopy slides. After deparaffinization and rehydration, the sections were stained in Weigert’s hematoxylin for 10 min, washed again with tap water for 1 min, and rinsed in distilled water. Next, the slides were stained with Biebrich scarlet-acid fuchsin for 10 min, rinsed in distilled water, incubated with phosphotungstic-phosphomolybdic acid for 5 min, stained with aniline blue for 10 min, and fixed in 1% acetic acid for 2 min. Finally, the slides were rinsed in distilled water, dehydrated, and mounted.

Fibrous tissue was stained blue, the cytoplasm was stained red, and cell nuclei were stained violet. The collagen volume fraction (CVF) was quantified by calculating the percentage of area of collagen staining using Image-Pro Pro Plus v6.0 software. The CVF was calculated as the collagen area/total area. Five 200× fields from each sample were randomly chosen under a microscope, and the average CVF in the fields was used for analysis.

### 2.7. Statistical Analysis

All values are presented as the mean ± standard deviation. Statistical analysis was performed with SPSS 22.0 software (IBM, New York, NY, USA). One-way ANOVA followed by Tukey’s post hoc test was used to analyze differences between groups, and *p* < 0.05 was considered statistically significant. Masson’s trichrome-stained sections were observed and photographed under a light microscope (BX51 OLYMPUS), and the CVF was measured with Image-Pro Plus 6.0 software. Histograms were generated with GraphPad Prism 5.01 software.

## 3. Results

### 3.1. The ALCAT1 Expression Was Increased in MI Rats

It is well known that ALCAT1 plays a major role in CL metabolism in some diseases. We sought to determine whether ALCAT1 is metabolically active in acute myocardial infarction. As illustrated in Figure 2, compared with that in the sham group, ALCAT1 expression in the heart was increased after MI for 1 h–5 days. However, in serum, the results showed that the ALCAT1 protein content in the MI-1 h group and the MI-6 h group was significantly increased compared with that in the S group. These results indicated that MI resulted in significantly increased myocardial and serum ALCAT1 expression in the rats, and serum ALCAT1 after MI for 1–6 h can be used as a marker for the early diagnosis of myocardial infarction.

### 3.2. Myocardial Injury Was Distinct in MI Rats

To determine whether MI caused myocardial injury, kits were used to detect markers of myocardial injury parameters in the serum of all the rats. As illustrated in Figure 3, we found that, compared with group S, CK activity showed an upward trend at 1 h after MI, while it increased significantly and continuously through 6 h, peaked at 24 h, and returned to normal levels 5 days after MI. In addition, after MI for 6 h, CKMB activity began to increase and increased significantly at 12 h and 24 h, after five days, it returned to a normal level. Serum LDH activity was significantly increased from 6 h to 5 days after MI and the apparent rise in BNP, a factor that reflects infarct size and left ventricular function, was an hour after MI. These results suggest that MI leads to severe myocardial injury in rats. The effect of serum CK activity and CKMB activity was significant at 24 h after MI, while the effect of LDH activity was not limited to 24 h; it exerted an effect five days after MI, which also indicated that the characteristics of myocardial injury caused by MI were sustained.

### 3.3. Myocardial Fibrosis Was Distinct in MI Rats

To evaluate the effect of MI on myocardial fibrosis, Masson trichrome staining was performed in all groups. As shown in Figure 4, the collagen fibers were blue, the cardiomyocytes were red, and the nuclei were blue violet. CVF was significantly increased in the MI-5d group compared with the S group. No myocardial infarction was observed in control or sham-operated hearts. Fortunately, five days after myocardial infarction, the myocardial infarct size was significantly increased compared with that in the sham rats. These data indicated that myocardial fibrosis occurred five days after MI.

### 3.4. Myocardial Function Was Impaired in MI Rats

To evaluate the effect of MI on cardiac hemodynamics, ventricular function was assessed in all groups. There was a significant reduction in LVSP, +dp/dt max and −dp/dt max in the MI group compared with the S group (Figure 5), while LVEDP was significantly increased in the MI group. Together, these data showed that cardiac function was impaired after MI.

### 3.5. Exercise Training Mitigated Cardiac Dysfunction after MI

Then, we used Masson’s trichrome staining to determine the degree of myocardial fibrosis. We did not observe myocardial infarction in control or sham-operated rats. Surgery resulted in significant myocardial infarction in both sedentary and exercising rats compared with rats in the corresponding sham groups, as evidenced by a large blue-stained area of myocardial fibrosis. Notably, the myocardial fibrosis was significantly decreased in the rats subjected to four weeks of exercise training compared with sedentary rats. The collagen volume of fraction (CVF %) in heart sections, which was quantified with Masson’s staining (Figure 6A,B), was significantly decreased by exercise in MI rats.

Next, we further asked whether exercise alleviates MI-induced cardiac dysfunction. ELISA kits showed that exercise training protected against MI-induced myocardial injury (Appendix A). Similarly, cardiac function analysis revealed that MI also induced remarkable dysfunction of the hearts of sedentary rats. An increase in LVEDP is characteristic of diastolic dysfunction post-MI, and exercise training maintained this parameter at normal levels, as shown in Figure 6C–F. This finding, along with the reduction in collagen deposition, suggests that exercise training mitigated cardiac dysfunction in the MI + EX group.

### 3.6. Exercise Training Decreased the Expression of ALCAT1 Following MI

ALCAT1 is a pivotal mediator of CL metabolism and mitochondrial function and has a remodeling effect on CL branch chain structure. CL remodeling mediated by ALCAT1 can accelerate reactive oxygen production and mitochondrial damage, leading to apoptosis. Thus, to explore the underlying role of ALCAT1 in exercise-induced cardioprotection, we performed RT-qPCR and Western blotting. As illustrated in Figure 7A–C, a marked increase in ALCAT1 expression was observed in the MI group, and ALCAT1 expression was decreased in the MI + EX group compared with the MI group, showing that exercise training attenuates the increase in ALCAT1 expression in both the myocardium and serum. Taken together, these results suggest that ALCAT1 is a key player in exercise-induced cardioprotection.

### 3.7. Exercise Training Increased Myocardial Antioxidant Activity Following MI

It has been well demonstrated that myocardial functional changes following MI are associated with increased oxidative stress and lipid peroxidation. Thus, we measured T-AOC, GSH-PX, CAT, SOD, and MDA levels in the serum and heart, respectively. As shown in Figure 8, serum and heart oxidase activity and lipid peroxidation levels were obviously increased after MI, and decreases in total antioxidant capacity and antioxidant enzyme activity were observed in the MI group. However, there was substantial improvement in antioxidant capacity at the end of the four-week exercise period, and interestingly, MI + EX partially reversed the increase in lipoperoxidation, as determined by MDA levels and shown in Figure 8J,K. Moreover, we used Western blotting to examine myocardial SOD1 and SOD2 expression in rats receiving different treatments. As shown in Figure 8I, the SOD1 and SOD2 levels were slightly increased in the myocardium in rats subjected to four weeks of aerobic exercise compared with sedentary CTL and MI rats, suggesting that aerobic exercise protects the heart from MI-induced myocardial injury by elevating antioxidant levels and mitigating the production of lipid peroxidation products.

### 3.8. Exercise Training Inhibited MI-Induced Cardiomyocyte Apoptosis

We further used Western blotting and caspase-3 activity assays to test whether inhibition of apoptosis is linked to exercise-induced cardioprotection. The results showed that the expression of apoptotic proteins (Cyt C and P53) was confirmedly decreased and that the ratio of Bcl-2 and Bax was markedly increased in the MI group compared with the MI + EX group, as shown in Figure 9. Similar to this MI-induced increase in the levels of apoptotic proteins, analysis of caspase-3 activity revealed that MI also induced a significant rise in caspase-3 activity in the hearts of sedentary rats. However, exercise training attenuated this MI-induced increase in myocardial caspase-3 activity. Collectively, these findings, together with the proteinic and enzymatic changes, indicate that exercise training alleviates MI-induced myocardial dysfunction by restraining MI-induced cardiomyocyte apoptosis.

## 4. Discussion

The connection of CL with many proteins and inner and outer membranes provides the required electrochemical gradient for ATP generation. CL plays a very important role in cell oxyacid phosphorylation and ATP synthesis, maintaining mitochondrial function and promoting cell survival. As an important phospholipid molecule, CL regulates mitochondrial energy generation. ALCAT1 plays a key role in CL peroxidation and is an acyltransferase. Studies have found that ALCAT1 is located on the connective membrane between the endoplasmic reticulum and mitochondria and is abundant in the myocardium and liver. The findings were further supported by isolated liver and heart microsomes. It was shown that the ER has a regulatory effect on phospholipid metabolism [18,27,28,29,30,31]. The heart is very sensitive to ischemic and hypoxic signaling. When acute myocardial infarction occurs, a large number of ROS are produced after myocardial ischemia and hypoxia, resulting in abnormal metabolism of myocardial tissue [7,9,12,32]. The dynamic balance between the heart and the circulatory system is broken. This study showed that lclat1mRNA expression increased significantly 6 h, 12 h, 24 h, and 5 d after myocardial infarction and that the ALCAT1 protein content significantly increased 1 h, 6 h, 12 h, 24 h, and 5 d after myocardial infarction. Serum ALCAT1 protein expression was significantly increased 1 h and 6 h after MI. These results suggest that myocardial infarction may significantly increase the expression of ALCAT1 in the local myocardium and serum of rats. These results suggest that myocardial and serum ALCAT1 expression is significantly increased in myocardial infarction rats, and serum ALCAT1 can be used as a marker for the early diagnosis of myocardial infarction from 1 to 6 h after MI.

When acute myocardial infarction occurs, a large number of myocardial cells are damaged, and apoptosis results from ischemia and hypoxia caused by blocked cardiac blood circulation, which seriously affects the recovery of myocardial infarct tissue. Recent studies have found that when acute myocardial infarction occurs, myocardial endothelial cell and mitochondrial damage worsen and lead to apoptosis and myocyte necrosis, which can significantly reduce the function of endothelial cells, resulting in considerable endothelial cell damage and a high apoptosis rate [33,34,35], eventually causing cardiac dysfunction and heart failure. After AMI, the necrotic myocardium releases enzymes into the blood that can be measured. The release of myocyte-damage-related myocardial enzymes, such as CK, CKMB, and LDH, is increased. The content of myocardial enzyme activity in the myocardium and circulation can be used as a standard for detecting myocardial infarction injury. The results of this study showed that the detection results of the rat serum kit indicated that CK showed an upward trend 1 h after myocardial infarction, significantly increased 6 h after infarction, and continued to increase. The expression level reached its highest at 24 h and returned to normal after 5 d. CKMB began to increase 6 h after myocardial infarction, increased significantly at 12 h and 24 h, and returned to normal levels after five days. Serum LDH was continuously elevated from 6 h after MI. In conclusion, the expression levels of CK, CKMB, and LDH in the serum of rats after myocardial infarction increased significantly from 6 to 12 h and reached the highest level at 24 h, while LDH continued to increase at five days, indicating that myocardial injury occurred after myocardial infarction.

Among many cardiovascular diseases, AMI is the leading cause of high hospitalization and mortality rates. A large number of studies and clinical data have shown that acute myocardial infarction can seriously damage cardiac function, with significant changes in left ventricular diameter and ejection fraction, resulting in cardiac fibrosis, ventricular remodeling, and reduced cardiac function [36,37,38]. After MI, remodeling of the left ventricle leads to fibrotic scarring of the cardiovascular system, blocking the normal circulation of blood, leading to changes in the load on the heart before and after MI and ultimately to heart failure [39,40]. Studies have shown that 21 days after acute myocardial infarction in mice, many myocardial cells underwent apoptosis, the levels of surrogate markers of fibrosis increased in the myocardial infarction area, and cardiac function significantly decreased [41]. It has been suggested that acute myocardial infarction can seriously reduce cardiac function and harm the heart. The results of this study showed that after acute myocardial infarction, the cardiac function of rats declined severely, the number of myocardial collagen fibers increased significantly five days after myocardial infarction, and the myocardium exhibited compensatory hypertrophy.

Myocardial ischemia and hypoxia after MI initially lead to inflammation and oxidative stress injury [26,42,43,44], which are followed by cardiomyocyte apoptosis and fibrosis [26,45] and ultimately cardiac dysfunction [23,46,47,48]. The current clinical treatments for MI are reperfusion therapy, β-receptor blocker drug therapy, and antiplatelet therapy. Unfortunately, myocardial infarction cannot be effectively treated due to various limitations, such as the timing of treatment, the interval to reperfusion, drug resistance, and the cost of treatment. Therefore, it is urgent to identify economical medical treatments with fewer side effects for treating cardiovascular diseases. Exercise training, in addition to reducing cardiovascular risk factors, confers direct protection against MI injury and has been associated with improved heart attack survival in humans [5,49,50]. The hemodynamic data confirmed that left ventricular systolic dysfunction and remodeling occurred in MI rats, and exercise reduced systolic dysfunction, improving left ventricular systolic function, confirming the findings of our previous study [6,7,8,9,10]. Relevant studies have confirmed that exercise can improve cardiac function after myocardial infarction [51,52], and the improvement in cardiac function is more obvious when exercise rehabilitation is performed earlier [53]. The literature has shown that appropriate exercise reduces myocardial oxidative stress [54,55,56], inhibits cardiomyocyte apoptosis [55], decreases myocardial fibrosis, and improves myocardial interstitial structure in heart failure rats [6,7,8,9,10]. Eleven weeks of aerobic exercise can improve the aortic antioxidant capacity of myocardial infarction rats and alleviate the effects of myocardial infarction [57]. Furthermore, our published findings reported in 2021, showed that, in the myocardium of MI mice, aerobic exercise reduced oxidative stress [26]. Interestingly, in this study, it was found that, in rats, the activity of antioxidant enzymes in both the myocardium and circulation were significantly decreased and the level of lipid peroxidation was significantly increased after myocardial infarction, while after four weeks of aerobic exercise, these indicators tended to be normal. These are consistent with the abovementioned findings reported in the literature. Our results conclude that exercise can mitigate oxidative stress in the heart and circulation of MI rats.

After myocardial infarction, apoptosis of myocardial cells decreases significantly [55]. Evidence has confirmed that exercise has a protective effect on the myocardium by resisting myocardial ischemia-induced apoptosis through the myocardial mitochondrial pathway [58,59]. It has been verified that long-term exercise can markedly increase the expression of bcl-2 mRNA in cardiomyocytes and inhibit the apoptosis of cardiomyocytes [17]. Pathological remodeling of CL is a key contributor to mitochondrial structure and functional impairment caused by chronic noncommunicable diseases, such as obesity, diabetes, and pathological cardiac hypertrophy [19,20]. Moreover, ALCAT1 is a key player in CL metabolism. ALCAT1 accelerates the pathological remodeling of CL by ROS, resulting in a decrease in CL content and further leading to mitochondria-induced apoptosis [16,17,19,20,21]. Studies have shown that ROS impair mitochondrial structure and function by oxidizing CL [18]. CL oxidation is accompanied by the release of Cyt C, which then initiates the caspase cascade pathway to induce cell apoptosis [16]. It has been reported that ALCAT1-mediated CL pathological remodeling can accelerate CL oxidation and reduce CL content in models of myocardial hypertrophy, heart failure, diabetes, aging, obesity, and other diseases [16,17,19,20,21]. Moreover, studies in H9C2 cells have found that when expressed at high levels, ALCAT1 is very sensitive to ROS, and a large amount of ROS will oxidize more lipids. ROS also induce ALCAT1 overexpression, and heart tissues and muscles are more sensitive than other tissues to this effect [60]. Therefore, inhibiting pathological CL remodeling is an effective strategy to prevent and treat chronic noncommunicable diseases [23]. However, there have been no reports on this topic. Recently, increasing attention has been given to the effect of eight weeks of aerobic exercise, which can increase the proportion of CL in the liver of high-fat diet-fed rats; however, low-intensity aerobic exercise can significantly increase the content of CL in the myocardium of rats with spontaneous hypertensive heart failure [23,24].

It was previously reported that, in MI mice, the effect of six-week aerobic exercise on cardiomyocyte apoptosis was better than that of no training [26]. Coincidentally, we also found this phenomenon in MI rats after four weeks of aerobic exercise. Myocardial Bcl-2 protein expression and the Bcl-2/Bax ratio increased significantly and the protein expression of Cyt C, P53, and Bax and the activity of caspase 3 markedly decreased, after four weeks of aerobic exercise. Similarly, there was a sharp improvement in cardiac function and a great reduction in the myocardial collagen percentage. The results showed that aerobic exercise could effectively improve cardiac function by downregulating ALCAT1 protein expression and inhibiting cell apoptosis in MI rats.

## 5. Conclusions

Taken together, the above data demonstrated that the potential utility of MI leads to severe heart injury in rats and increases the expression of ALCAT1 in the rat myocardium and serum. The serum ALCAT1 index can be used as a marker for the early diagnosis of myocardial infarction, within 1–6 h of MI. Serum CK and CKMB had significant detection effects in myocardial infarction at 24 h, while the LDH index was not limited to a 24 h change and exerted an effect five days after MI, which indicates that myocardial injury caused by myocardial infarction continuously develops and that myocardial replacement fibrosis occurs five days after myocardial infarction in rats. Four-week aerobic exercise improved cardiac function and pathological remodeling through decreasing ALCAT1 expression and oxidative stress level in myocardium and serum, and restraining cardiomyocyte apoptosis, as illustrated in Figure 10. This study provides what we believe is novel insight into the mechanism underlying exercise-induced cardiac repair. ALCAT1 may be considered as a potential marker of early MI disease. All these findings may eventually shed light on effective nonpharmacological and pharmacological exercise-based strategies for the management and rehabilitation of MI patients.

## Figures and Tables

**Figure 1 biomedicines-10-02250-f001:**
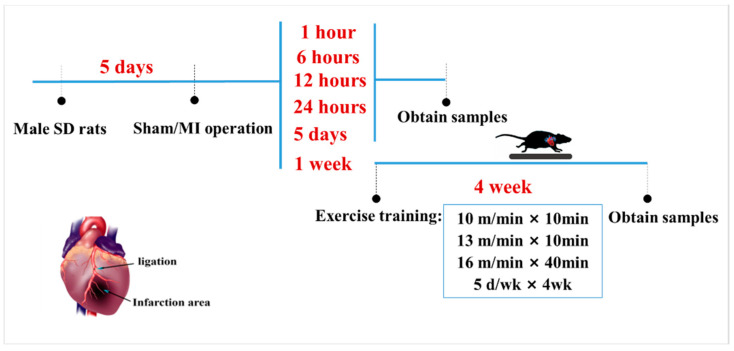
Schematic diagram of the animal experimental design and protocol. One week after the operation, male Sprague-Dawley rats in the MI + EX group and EX group were subjected to exercise on a motorized rodent treadmill for 4 weeks. All rats were euthanized for cardiac tissue and serum sample collection after a 4-week exercise protocol.

**Figure 2 biomedicines-10-02250-f002:**
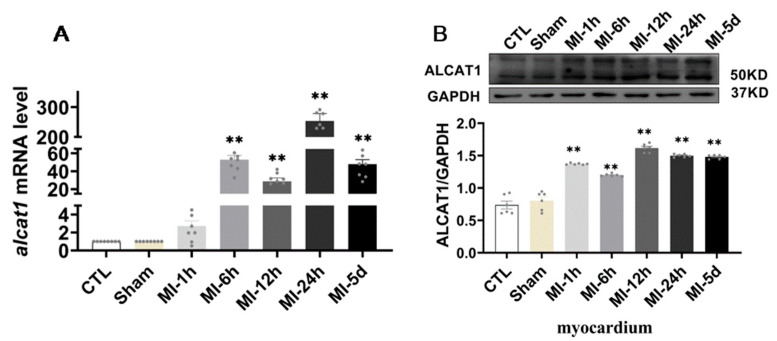
The expression of ALCAT1 mRNA and protein. (**A**) Quantitative reverse transcription polymerase chain reaction (RTq–PCR) analysis of ALCAT1 in the heart (*n* = 3). (**B**) Western blot images and densitometric quantitative analysis of ALCAT1 protein expression in the heart (*n* = 4). Values presented are the means ± SEM. CTL, normal control; sham; MI-1 h, 1 h after myocardial infarction; MI-6 h, 6 h after myocardial infarction; MI-12 h, 12 h after myocardial infarction; MI-24 h, 24 h after myocardial infarction; MI-5d, 5 days after myocardial infarction (** *p* < 0.01 vs. sham).

**Figure 3 biomedicines-10-02250-f003:**
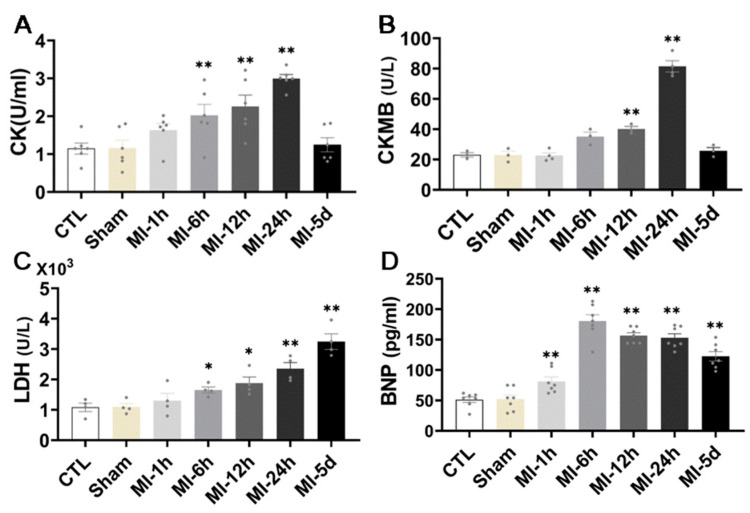
Effect of MI on myocardial injury. ELISA kit analysis of myocardial injury level, CK (**A**), CKMB (**B**), LDH and BNP (**C**,**D**) in rat serum (*n* = 6). Values presented are the means ± SEM. CTL, normal control; sham; MI-1 h, 1 h after myocardial infarction; MI-6 h, 6 h after myocardial infarction; MI-12 h, 12 h after myocardial infarction; MI-24 h, 24 h after myocardial infarction; MI-5d, 5 days after myocardial infarction (* *p* < 0.05, ** *p* < 0.01 vs. sham).

**Figure 4 biomedicines-10-02250-f004:**
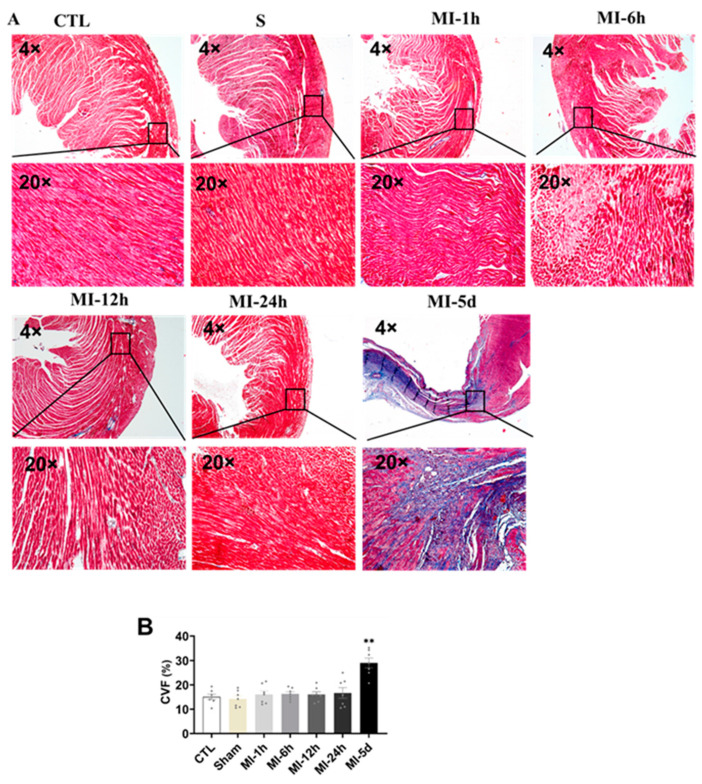
Tests of myocardial fibrosis after MI. Blue collagen, a fibrosis marker, red cardiac muscle fibers, and nuclei in dark brown emerged in the microscopic images. (**A**) Masson-negative staining portions (cardiac muscle fibers) and Masson-positive staining portions (blue collagen) were digitally measured by collagen volume of fraction (CVF %) (*n* = 3). (**B**) Values presented are the means ± SEM. CTL, normal control; sham; MI-1 h, 1 h after MI; MI-6 h, 6 h after MI; MI-12 h, 12 h after MI; MI-24 h, 24 h after MI; MI-5d, 5 days after MI (** *p* < 0.01 vs. sham).

**Figure 5 biomedicines-10-02250-f005:**
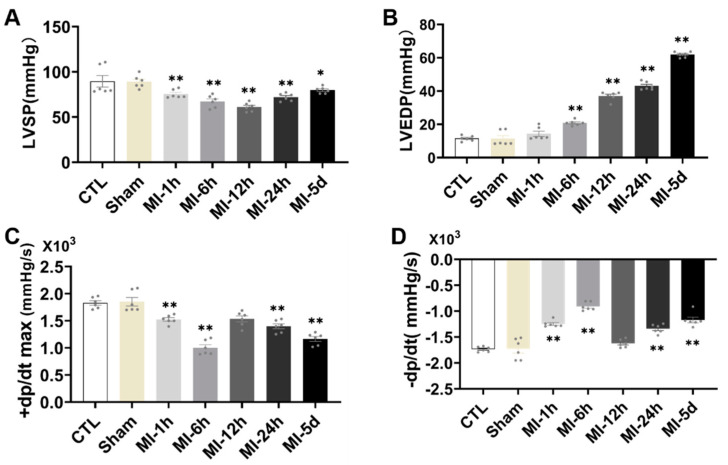
Tests of myocardial function after MI. The cardiac hemodynamic parameters measured included systolic and diastolic function of the left ventricle measured with (**A**) LVSP, (**B**) LVEDP, (**C**) +dp/dt, maximum pressure increasing rate, and (**D**) −dp/dt, maximum pressure decreasing rate (*n* = 6.7). Values presented are the means ± SEM. CTL, normal control; sham; MI-1 h, 1 h after myocardial infarction; MI-6 h, 6 h after myocardial infarction; MI-12 h, 12 h after myocardial infarction; MI-24 h, 24 h after myocardial infarction; MI-5d, 5 days after myocardial infarction (* *p* < 0.05, ** *p* < 0.01 vs. sham).

**Figure 6 biomedicines-10-02250-f006:**
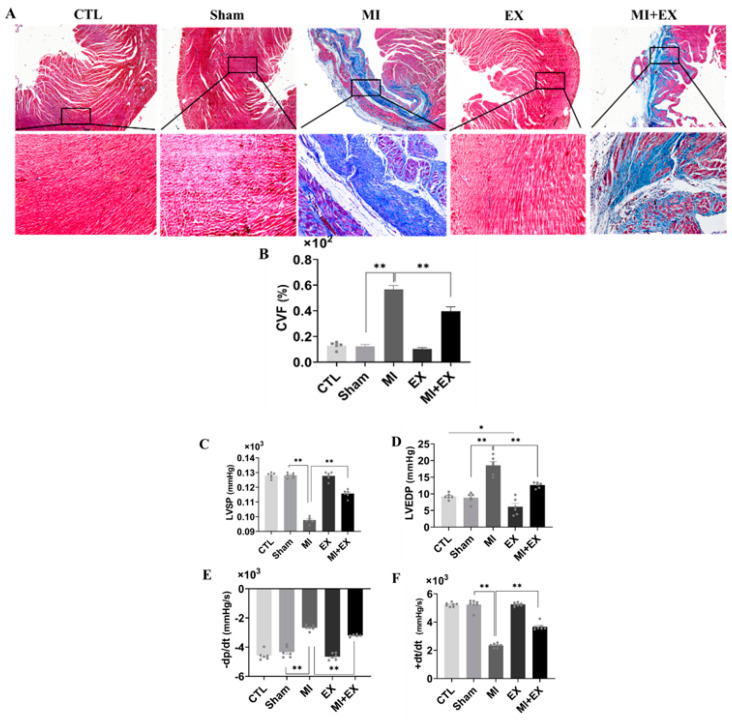
Exercise training mitigates cardiac dysfunction after MI. Blue-stained collagen, a fibrosis marker, red-stained cardiac muscle fibers, and dark brown nuclei visualized by microscopy (scale bar = 0.61 µm). (**A**,**B**) The areas of no Masson’s trichrome staining (cardiac muscle fibers) and Masson’s trichrome-positive staining (blue-stained collagen) were digitally measured to determine the collagen volume fraction (CVF %) (*n* = 3). The systolic and diastolic function of the left ventricle was assessed by measuring cardiac hemodynamic parameters including (**C**) LVSP, (**D**) LVEDP, (**E**) −dp/dt, maximum pressure decreasing rate, and (**F**) +dt/dt, maximum pressure increasing rate (*n* = 6.7). The values are presented as the mean ± SEM. CTL, normal control; MI, myocardial infarct; EX, CTL + exercise; MI + EX, MI + exercise. * *p* < 0.05, ** *p* < 0.01.

**Figure 7 biomedicines-10-02250-f007:**
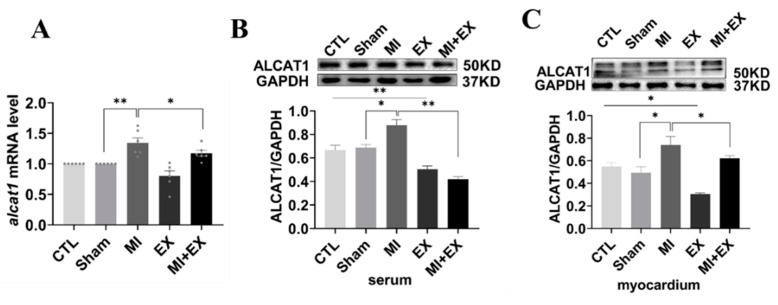
mRNA and protein expression of ALCAT1. (**A**) Quantitative reverse transcription polymerase chain reaction (RTq-PCR) analysis of ALCAT1 expression in the heart (*n* = 6). (**B**,**C**) Western blot images of ALCAT1 protein in the serum and heart and densitometric analysis of the bands (*n* = 3). The values are presented as the mean ± SEM. CTL, normal control; MI, myocardial infarct; EX, CTL + exercise; MI + EX, MI + exercise. * *p* < 0.05, ** *p* < 0.01.

**Figure 8 biomedicines-10-02250-f008:**
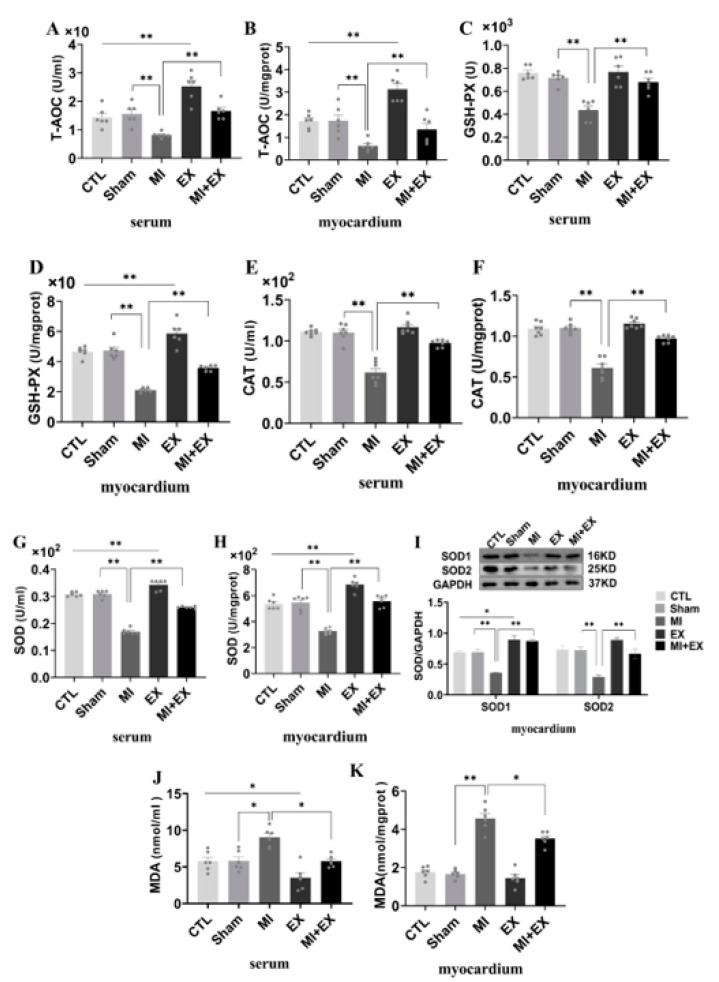
Effect of exercise and no exercise on myocardial oxidative stress following MI. ELISA of antioxidant levels. T-AOC (**A**,**B**), GSH-PX levels (**C**,**D**), CAT levels and SOD levels (**E**–**H**) in serum and tissue. (**I**) Representative blots of superoxide dismutase 1 and 2 (SOD1 and SOD2) and quantification of these data. (**J**,**K**) The level of MDA, an indicator of lipid peroxidation, was determined in rat serum and heart tissue samples with a TBARS kit. The values are presented as the mean ± SEM (*n* = 3, 6, 7). CTL, normal control; MI, myocardial infarct; EX, CTL + exercise; MI + EX, MI + exercise. * *p* < 0.05, ** *p* < 0.01.

**Figure 9 biomedicines-10-02250-f009:**
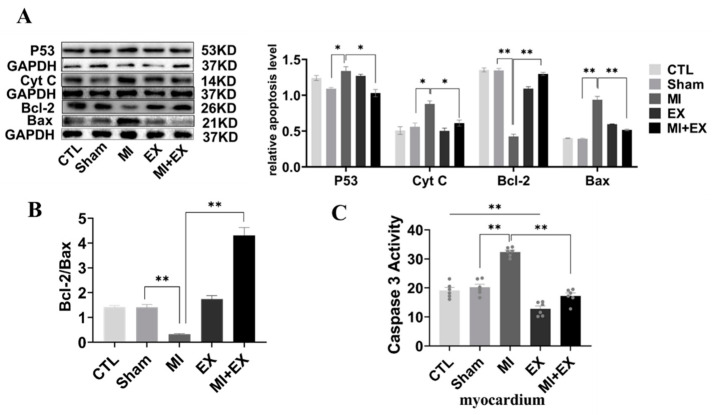
Effect of exercise and no exercise on cardiomyocyte apoptosis following MI. (**A**) Representative blots of the apoptotic proteins, p53, Cyt C, Bcl-2, and Bax, in tissue (*n* = 3). (**B**) Bcl-2/BAX ratio. (**C**) The activity of caspase 3 in the myocardium was measured using a spectrophotometer (*n* = 6). The values are presented as the mean ± SEM. CTL, normal control; MI, myocardial infarct; EX, CTL + exercise; MI + EX, MI + exercise. * *p* < 0.05, ** *p* < 0.01.

**Figure 10 biomedicines-10-02250-f010:**
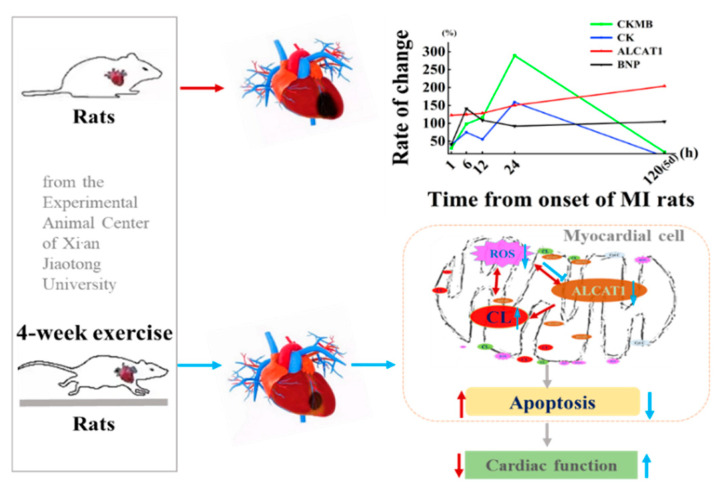
Proposed mechanism by which exercise-induced cardioprotection protects against myocardial infarction (MI)-induced injury in the heart. Exercise training inhibits the production of ALCAT1 in circulation and cardiac tissues and exerts cardioprotective effects against MI-induced injury by suppressing myocardial apoptosis and enhancing antioxidant activity.

## Data Availability

Data sharing not applicable. The data are not publicly available due to privacy restriction.

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
