# Peer review of "Cardioprotection Attributed to Aerobic Exercise-Mediated Inhibition of ALCAT1 and Oxidative Stress-Induced Apoptosis in MI Rats"

_biomedicines, 2022, doi:10.3390/biomedicines10092250_

Round 1

Reviewer 1 Report

The aim of the paper written by Niu Liu et al. is to consider cardioprotection attributed to aerobic exercise-mediated inhibition of ALCAT1 and oxidative stress-induced apoptosis in MI rats. They evaluated ALCAT1 expression and the possible impact of aerobic exercise training on ALCAT1 expression and its role in myocardial fibrosis, antioxidant capacity, and cardiomyocyte apoptosis in MI rats. They concluded that ALCAT1 damages the heart in MI and exercise training reduces the expression of ALCAT1, oxidative stress and myocardial apoptosis following MI-induced cardiac injury. Following are several suggestions/concerns aimed at improving the report:

1. With regards to the exercise protocol shown in Fig.1., I would like to ask whether there was a difference in oxidative stress between the different stages of exercise (i.e. across the five days)?

2. In Fig. 3, Are the markers of the effect of AMI on myocardial injury from the serum (Fig. 3), compatible to markers in the myocardium? This comparison might be valuable.

3. From Fig.4 the authors found obvious fibrosis changes in pathology after MI-5d. At MI-24 hr, pathology is unclear. At which point between these two, did fibrosis changes become evident?

4. In Fig.7., concerning mRNA and protein expression of ALCAT1, the results for EX and EX+MI in panel B (serum) and panel C (myocardium) are incompatible. How can this difference be explained?

5. In Fig. 9 the levels of bcl-2 and bax in panel A compared with bcl-2/bax result in panel B is not compatible for the EX and EX+MI groups. Furthermore, the lower level of Caspase 3 in the myocardium shown in panel C for EX seems significant. How can the effects of EX explain this

Reviewer 2 Report

This experimental study examines the role of exercise on cardioprotection after myocardial infarction (MI) and whether Acyl-CoA:lysocardiolipin acyltransferase 1 (ALCAT1) plays a role in MI effects. The study was conducted in rats subjected to MI followed by various exercise regimes. Various parameters of cardiac function and injury were assessed together with ultrastructural changes. It was shown that exercise was able to ameliorate the effects of MI, and that ALCAT1 appeared to play a role in this.

This is an interesting study that proposes a novel mechanism for amelioration of myocardial injury following an MI. The study has been well conducted and the manuscript has been well written. The manuscript may be slightly too long, but there has been significant work carried out and the results have been well described. There are some issues with the manuscript and these are highlighted below.

Specific Comments.

Abstract:

·           What is the relevance of cardiolipin in heart disease? Provide a rationale for conducting this study!

·           What is the role of ALCAT1 in MI?

·           Has it been shown that aerobic exercise alters ALCAT1 expression and thus has a role in protection induced by exercise?

·           Why is this novel when it states above that it plays a key role in heart diseases? Logic doesn't seem to add up!

Introduction:

·           Introduction gives a good background to the study.

Methods:

·           Fig 10 does not indicate that healthy male rats were obtained from etc etc! None of what is described in Section 2.1 is shown in Fig 10!

·           Line 84 and following - why are the groups repeated but now with 12 per group rather than 10 per group?

·           Line 94 - How was the heart exposed to induce a MI?

·           Line 116 - these are not cardiac function tests! What do the authors mean by 'cardiac function'?

·           Methods seem well described.

Results:

·           Lines 205-207 appear to be instructions to authors!

·           Line 218 - what is meant by 'kits were used to determine cardiac function'? How can function be determined by kits? Should this be cardiac metabolism or cardiac injury?

·           Line 228 - BNP shown in Fig 3 but not described in text!

·           Line 250 - Assessment of cardiac function (hemodynamics) has not been described in the Methods section! How was this done?

·           Why is -dp/dt in Fig 6 shown in a different format to that of Fig 5.

·           Results generally well described.

Discussion:

·           Describe the results in the context of other similar studies.

Round 2

Reviewer 1 Report

I appreciate the authors’ responses to my suggestions.

Author Response

We thank the reviewer for all helpful comments and suggestions, which have collaborated to improve the quality of our manuscript.

Reviewer 2 Report

The authors have responded well to the comments and suggestions made by this reviewer. The changes made to the manuscript have improved the manuscript and clarified the various issues that were commented on. Well done!

Author Response

(The authors gave the same response as above.)
